# Effects of Livestock Grazing on Spatio-Temporal Patterns and Behaviour of Reeves’s Pheasant *Syrmaticus reevesii*

**DOI:** 10.3390/ani12212968

**Published:** 2022-10-28

**Authors:** Junqin Hua, Shuai Lu, Kai Song, Jiayu Wang, Jinfeng Wang, Jiliang Xu

**Affiliations:** 1School of Ecology and Nature Conservation, Beijing Forestry University, Beijing 100083, China; 2Key Laboratory of Animal Ecology and Conservation Biology, Institute of Zoology, Chinese Academy of Sciences, Beijing 100101, China

**Keywords:** anthropogenic disturbances, sympatric distribution, activity rhythm, adaptation, nature reserve, occupancy model

## Abstract

**Simple Summary:**

Previous studies have confirmed the effects of livestock grazing on the habitat use of ground-nesting birds, while little is known about the effects of livestock on their behaviour. In this study, we used camera traps to investigate the effects of livestock presence on the spatio-temporal patterns and behaviour of Reeves’s Pheasant. We found that livestock presence altered the behavioural patterns of Reeves’s Pheasant. The findings suggest the need to incorporate behavioural impacts into wildlife conservation.

**Abstract:**

Protected areas are seeing an increase in anthropogenic disturbances in the world. Previous studies have demonstrated the impact of livestock grazing and human presence on the habitat use of birds, whereas little is known about the effect of free-ranging livestock on bird behaviour. Reeves’s Pheasant (*Syrmaticus reevesii*) is endemic to China and has been threatened by habitat loss and fragmentation, illegal logging, and human disturbance over the past 20 years. Based on camera trapping in the Liankangshan National Nature Reserve (LKS) and the Zhonghuashan Birds Provincial Nature Reserve (ZHS), we explore the effects of livestock grazing and human activities on the spatio-temporal distribution and behavioural patterns of Reeves’s Pheasant. Livestock does not appear to affect habitat use by the pheasant but changes its behavioural patterns. In addition, pheasants in areas with livestock foraged mostly during the early morning, while in areas without livestock, they foraged at dusk. Therefore, the study concludes that livestock intensity in nature reserves may have reduced pheasant suitability through altered patterns of vigilance and foraging behaviour.

## 1. Introduction

Globally, the rapidly growing livestock sector is one of the most important drivers of land-use change [1]. Livestock farming exacerbates habitat loss and degradation, pollution, invasive species spread, and disease transmission [1,2,3]. Moreover, livestock can compete with wildlife for limited food and space, thereby threatening their survival [4,5]. Understanding the interactions between livestock and wild animals is essential to improving human well-being and maintaining wild animal populations and habitats [6].

Although livestock grazing is often considered to be low-level human disturbance [7], the impacts of livestock grazing on wildlife are usually negative [8,9]. Recently, livestock in protected areas worldwide has increased, especially in forest landscapes, possibly affecting the predator abundance, breeding success, and habitats of wild animals [8,10,11]. Therefore, livestock can have a knock-on effect on ground-nesting and insectivorous birds that depend on forest resources [11,12,13].

To understand the effect of livestock grazing on wildlife, previous studies have compared the effect of different levels of livestock grazing intensity on the abundance of wildlife and resources at a spatial scale [11,12,14]. Camera traps can be useful for investigating the interactions between livestock and wildlife at a temporal scale [15]. Recently, camera trapping has increasingly provided a non-invasive way to detect elusive species [16]. A growing number of studies have used camera trapping data to study the effects of livestock on spatial and temporal patterns of wildlife [17,18]. In general, wild animals can shift their dietary activity patterns to effectively reduce competition with livestock [9,19]. For example, in the presence of livestock, Javan deer *Rusa timorensis* and wild boar *Sus scrofa* switch from a diurnal to a nocturnal activity pattern [17]. In fact, livestock can influence not only species richness and occupancy probability [12,14,20] but also the daily activity patterns and behaviour of animals, particularly foraging and anti-predatory behaviour [9,21].

Reeves’s Pheasant is a large terrestrial forest bird that was once widely distributed in the mountains of central China. A recent study has shown that pheasant has disappeared from 46% of the sites surveyed and its population size is declining throughout most of their distribution [22]. Reeves’s Pheasant is timid and difficult to observe, but its population dynamics are a barometer of habitat change [23]. Over the last 20 years, its habitats have been at risk of degradation and fragmentation [24]. There is a possible negative correlation between livestock grazing and the population size of Reeves’s Pheasant [25]. Given the ever-increasing contact zone with livestock, it is important to understand how Reeves’s Pheasant responds to livestock impacts and the mechanisms that govern those responses, which is important in conservation planning [26]. However, few studies have evaluated the impact of human disturbance on Reeves’s Pheasant at a fine spatial scale (<1 km) [22,27], and little information is available on the behaviour of Reeves’s Pheasant in the wild. 

The aim of our study was to provide science-based conservation strategies for a regional landscape-scale recovery plan for Reeves’s Pheasant to resolve human-wildlife conflicts. We used 12-month camera trapping data from two nature reserves in central China to analyse the spatio-temporal distribution patterns and behaviour of Reeves’s Pheasant and the ecological impact of livestock. We explored three questions. First, do livestock influence the spatio-temporal distribution patterns of Reeves’s pheasant? Second, do livestock change the behavioural patterns of pheasants on a spatio-temporal scale? Third, are there seasonal differences in the impact of livestock on pheasants? We expect the impact of livestock on the spatio-temporal distribution patterns of Reeves’s pheasant to be more obvious during the breeding season, because incubating birds are bound to their ground nests during this period, and livestock presence raises pheasant vigilance and reduces their foraging.

## 2. Materials and Methods

### 2.1. Study Area

We conducted fieldwork in two nature reserves in the Dabie Mountains, China. The Liankangshan National Nature Reserve (114°45′–114°52′ E, 31°30′–31°42′ N, hereafter ‘LKS’) is a national nature reserve (105.80 km^2^) in Xin County, Henan province (Figure 1) with over 600 wild Reeves’s Pheasants [22]. The reserve contains broadleaved forests and coniferous mixed forests at elevations of 100 to 700 m. The Zhonghuashan Birds Provincial Nature Reserve (113°54′–113°59′ E, 31°37′–31°44′ N, hereafter ‘ZHS’) is located in Guangshui City, Hubei province (Figure 1). ZHS is one of the smallest provincial nature reserves (78.20 km^2^) with over 800 wild Reeves’s Pheasants [22]. The reserve contains subtropical evergreen broad-leaved forests and deciduous broad-leaved mixed forests at elevations of 150 to 810 m. LKS and ZHS are located approximately 100 km apart, sharing a similar latitude and climate. 

### 2.2. Camera Trapping

We deployed camera traps (Ltl Acorn 6310 WMC, Zhuhai, China) at 35 sites in LKS and 37 sites in ZHS from July 2018 to July 2019 (Figure 1). Based on previous line transects conducted in the two nature reserves [28], these camera traps covered a large part of the suitable habitat for Reeves’s Pheasant. Camera traps were installed on animal trails at a height of 30 cm in the forest. The distance between any two camera traps was greater than 0.4 km [27]. Latitude and longitude coordinates, vegetation, and altitude of each camera trap station were recorded. Cameras were set to operate 24 h per day and were programmed to take up to 3 photos once triggered, followed by a 10 s video, with a 30 s interval to trigger the next photograph (Appendix A). Batteries and memory cards were replaced every 3 months. Camera traps were replaced if lost.

For each photo and video taken by camera traps, the location (camera number), date, and time were recorded. We identified and recorded each captured entity as an animal species, human, or livestock. Image records were screened to identify ‘independent photographic events’ based on a temporal separation criterion of more than 0.5 h between consecutive images of the same species [16,29] to avoid repeated counting of a single individual during a transitory stay close to the camera trap [29]. We divided camera trapping data into two periods, the breeding season (March to July) and the non-breeding season [30]. We collected data from 31 and 28 camera traps during the breeding season and non-breeding season, respectively, in LKS, and from 36 and 32 camera traps during the breeding season and non-breeding season, respectively, in ZHS. We calculated camera trap days (CTDs, the actual number of days each camera worked during the survey) as the average CTDs and standard error of the camera trap in each reserve and compared the average CTDs of each reserve during the breeding season, non-breeding season, and throughout the whole year using two-tailed unpaired two-sample Wilcoxon tests.

### 2.3. Environmental Variables

We tested a set of natural (including vegetation type, elevation, and distance from the nearest river) and anthropogenic (including distance from the nearest road, distance from the nearest settlement, distance from the nearest cultivated land, livestock encounter rates, and human encounter rates) variables that could affect the distribution of Reeves’s Pheasant [25,27,28,31]. The variables listed were assessed at the respective camera-trap locations either directly or downloaded from loaded open access sources (Table 1). Before the installation of camera traps, we conducted forest surveys and human footprint surveys in the study area and surrounding area. In addition to the coordinates of wildlife activity and human presence, we recorded the locations of human disturbances (including resident settlement, impervious roads, and agricultural land) and rivers, using portable GPS receivers (eXplorist 610). We measured the distance from each camera trap station to the nearest resident settlement, impervious road, river, and cultivated land using ArcGIS 10.4. Habitat types around camera trap stations were classified as broadleaf forests, conifer-broadleaf mixed forests, and coniferous forests. The elevation of each camera trap station was also recorded by a portable GPS receiver. Livestock encounters at each camera trap station were expressed as the number of detections per 100 CTDs [16,29].

We calculated the variance inflation factor (VIF) for all covariates to measure multicollinearity among these variables, and variables with a VIF < 5 were retained in the model (Appendix A). Furthermore, Pearson’s correlation coefficients were used to check for collinearity, and variables with *r* > 0.7 were not used in the model [32]. All covariates were retained as no collinearity was detected (VIF < 5.0 and *r* < 0.7) (Appendix A and Appendix A).

### 2.4. Data Analysis

#### 2.4.1. Occupancy Models

To analyse the habitat use of Reeves’s Pheasant during the breeding and non-breeding seasons, we fit single-species single-season occupancy models using the unmarked package [33]. These models estimated the occupancy probability (*ψ*) and detection probability (*p*) of a species for each camera trap station [34]. When the home range of a target species is larger than the effective monitoring area, the parameter *ψ*, usually referred to as occupancy probability, can be more accurately interpreted as the probability that the species use the area [18,35]. The detection probability (*p*) is estimated using the pattern of repeated visits of a target species to a site. The time frame visit occurrences are defined as a sampling occasion [36]. For this analysis, the sampling occasions were grouped by 3 days, and we divided the breeding season into up to 50 successive sampling occasions and the non-breeding season into up to 72 successive sampling occasions.

We evaluated the effects of natural and anthropogenic variables on *ψ* and *p*. We evaluated the effect of eight variables on the *ψ* of Reeves’s Pheasant. To assess the impact of human activity on *ψ*, we measured the distance from each camera trap site to the nearest settlement, road, and agricultural land. We predicted that Reeves’s Pheasant would avoid settlements and roads to avoid humans, and it would stay close to agricultural land for foraging. To assess the effect of natural environmental factors on Reeves’s Pheasant, we assessed the effect of distance from the nearest river, vegetation type, and elevation on *ψ*. Rivers are an important resource for wildlife to maintain normal activities, and differences in vegetation type and elevation have been reported to have important effects on the habitat use of target species [27,28]. Therefore, we predicted that *ψ* would decrease with increasing distance from water. In addition to the eight variables described above, we selected CTDs and the nature reserve where cameras were located to assess their effect on *p*. We predicted that the effect of natural and anthropogenic variables on *p* would be similar to that on *ψ*. We also predicted that CTDs and different study sites would have no effect on *p*, suggesting compliance with the closure hypothesis.

Models were ranked based on the Akaike Information Criterion (AIC) and all models with ΔAIC ≤ 2 were considered equally plausible [37]. We evaluated the relative importance of variables and computed average predictions and 95% confidence intervals using the MuMIn package (version 1.43.17) [38]. We created capture histories of Reeves’s Pheasant and livestock by collapsing individual days into 3-day periods to improve temporal independence and model convergence [39,40]. From each simulation, we calculated the sum of squared residuals (SSRs) as a measure of the model fit and compared the distribution of 1000 expected SSEs drawn from the simulation with observed SSEs calculated from fitting the predictions of the global model to our observations. A significant lack of fit would be indicated by observed SSRs being greater than 95% (i.e., *p* < 0.05) of the 1000 simulated values.

#### 2.4.2. Influence of Livestock on the Temporal Pattern of Reeves’s Pheasant Activity and Behaviour

We used the date and time of each Reeves’s Pheasant and livestock detection event to assess the temporal patterns of Reeves’s Pheasant–livestock interactions during the breeding and non-breeding seasons. Random samples from the continuous temporal distribution reflected the probability of the events being recorded at any interval during the day [41]. Based on the time of the independent photographic events, we estimated the diel activity pattern of each behaviour and individual in different seasons using kernel density estimation according to the approach proposed by Ridout and Linkie [42] with the “overlap” package (version 0.3.3) in R software [43]. Then, we used the coefficient of overlap, Δ, ranging from 0 (no overlap) to 1 (complete overlap), to measure the overlap between the distribution of Reeves’s Pheasant and livestock. The coefficient was obtained from the area under the curve of the minimum of two density functions at each time point. We estimated Δ using the Δ4 method because all samples were greater than 75 [44]. The 95 % confidence interval was obtained using 1000 bootstrap samples. Statistical analyses were performed using the overlap package, which provides a way to describe the overlap in activity rhythms while not providing a threshold to verify the differences in activity rhythms between species. We used the activity package (version 1.3.1) to verify whether two sets of circular observations came from the same distribution.

#### 2.4.3. Influence of Livestock on Spatial of Reeves’s Pheasant Behaviour

During surveys in LKS and ZHS, we analysed the behaviour of Reeves’s Pheasant. Behavioural analysis was conducted for each individual per event (3 photos, 10 s video) if an adult individual was clearly visible. For each detection event (16 s), we used the sampling method of ‘‘behaviour sampling’’ and the recording method of time sampling and one-zero sampling [45,46]. All Reeves’s Pheasants were observed as one group and each behavioural element (Appendix A) was marked as 1 if it occurred and 0 if it did not occur during a sampling interval. This method is increasingly popular in wildlife behaviour studies, especially studies with sampling periods of less than 20 s [45,46]. Such objective scoring is easy to achieve, but at the cost of key behavioural information discarding [45]. For example, such scores will not accurately indicate the frequency or proportion of time spent on a behaviour [45]. Instead, the scores for the various behaviours in this study can be interpreted as the proportional chance of these behaviours occurring on a given occasion.

We focused on the effects of livestock on the foraging, vigilance, and locomotion behaviour of Reeves’s Pheasant as these three behaviours are often used as a proxy for risk perception [47]. We tested the effects of the nine variables described above (excluding CTDs) on Reeves’s Pheasant’s vigilance (proportion of vigilance behaviours per camera point) using GLMMs with binomial distributions and logit link functions. Considering the spatial (camera traps station and study area) non-independence of sites, camera ID (nested within the study area) was added as a random factor. The best models were selected according to AIC values, and model averaging was performed with the dredge function of the MuMIn package. However, the distribution of these data did not lend themselves to any of the available generalized linear models and links [48], and the discrepancy in sample sizes between study areas did not lend the data to linear mixed-effect models [49]. After inspecting the data, we formulated a tentative hypothesis that these data were represented by two distributions in the dataset, indicating two linear relationships between response and predictor variables. In plain terms, we suspect that livestock activity influences behavioural variation in pheasants, and that the insufficient sample of high livestock activity stations does not capture this relationship. We tested this ad hoc hypothesis with a piecewise (or segmented) regression analysis, in the R package SiZer [50]. Using the same method, we analysed the effects of livestock and other covariates on the locomotion and foraging behaviour of Reeves’s Pheasant. We used Mann–Whitney U tests to compare the mean occurrence probabilities of the three main behaviours in the presence or absence of livestock. All statistical analyses were carried out in R (version 4.0.5) [43]. For all statistical tests, *p* < 0.05 was considered significant. 

Data collected from 69 camera trap stations yielded 18,263 CTDs and 705 independent photographic events of Reeves’s Pheasant and 702 independent photographic events of livestock (Appendix A). The average sampling effort per camera trap station over the entire survey period was 259.97 ± 90.53 CTDs in LKS and 268.76 ± 90.79 CTDs in ZHS. During the breeding season, we collected 114.16 ± 46.71 CTDs per camera trap station in LKS and 116.83 ± 49.52 CTDs per camera trap station in ZHS. During the non-breeding season, we collected 170.71 ± 50.60 CTDs per camera trap station in LKS and 179.31 ± 42.49 CTDs per camera trap station in ZHS. There were no significant differences in average CTDs between the two reserves during the breeding season, the non-breeding season or the whole year (Appendix A).

## 3. Results

### 3.1. Spatial Distribution of Reeves’s Pheasant and Livestock

During the breeding season, model averaging showed that Reeves’s Pheasant was more likely to be found in areas far from roads and close to settlements. The occupancy estimation model included both location and livestock encounter rates, but the 95% confidence interval for the estimated parameter (*β*) included 0 (Appendix A). There was no significant correlation between the occupation probability of the pheasant and the variables. During the non-breeding season, model averaging results included more parameters, and the occupancy probability in ZHS was higher than that in LKS. The detection probability estimation model included the vegetation type, elevation, distance from the nearest settlement, and livestock encounter rates, but none were significant. Distance from the nearest impermeable road is positively correlated with the detection probability of pheasants, while the distance from settlements is negatively correlated with its detection probability. Livestock activity did not appear to affect the spatial distribution of pheasants (Table 2).

### 3.2. Behavioural Patterns of Reeves’s Pheasant

When comparing the proportion of five behaviours observed in LKS and ZHS, we found that locomotion accounted for approximately 40% of the observations, followed by vigilance and foraging (Appendix A). The three behaviours related to risk perception accounted for more than 90% of the observations. Comfort behaviours and other behaviours accounted for less than 5% of the observations. The mean proportion of vigilance behaviour of the pheasant at sites with livestock absence was 28.88%, increasing to over 40% when livestock grazing intensity reached 30, as suggested by the piecewise linear model (Figure 2). The effect of livestock on pheasant vigilance behaviour appeared to diminish at livestock activity intensities greater than 30 (Figure 2). The probability of vigilance behaviour for Reeves’s Pheasant was significantly higher in areas with livestock than in areas without livestock (Mann–Whitney U test: U = 189.5, *n* = 59, *p* < 0.05) (Figure 3). Compared with livestock presence, the foraging behaviour and locomotion behaviour showed a downward trend in livestock absence. There was no significant effect of livestock on foraging and locomotion behaviour.

### 3.3. Temporal Overlap

There was a high degree of overlap between the daily activity patterns of birds and livestock, and there were significant differences in the timing of daily activity patterns throughout the year (mean Δ = 0.77 [0.71–0.79], *p* < 0.03) (Figure 4a). Reeves’s Pheasant tended to be more active in the early morning (6:30) and late afternoon (18:00), whereas the activity peak of livestock occurred around 15:00. When comparing the degree of overlap between the temporal distribution patterns of pheasants in areas where livestock were present and absent throughout the year (Figure 4b), we found high levels of overlap and no difference in the timing of daily activity patterns between two scenarios (mean Δ = 0.87 [0.83–0.94], *p* > 0.09). However, in areas where livestock were present, pheasants were more active in the morning. In contrast, in areas where livestock were absent, pheasants were more active at dusk. The general trend of the temporal distribution pattern of pheasants and livestock in the breeding season (mean Δ = 0.82 [0.74–0.86], *p* > 0.09) and non-breeding season (mean Δ = 0.70 [0.60–0.73], *p* < 0.001) was the same as that found over the whole year, but the degree of overlap between pheasants and livestock during the non-breeding season was relatively low (Figure 4d). Reeves’s Pheasant was more likely to forage in the morning in areas where livestock were present than in areas where livestock were absent (mean Δ =0.77 [0.66–0.88], *p* < 0.001, Figure 4e). We also found that Reeves’s Pheasant showed high vigilance throughout the day in areas where livestock were present, while they displayed high vigilance only in the morning and evening in areas where livestock were absent (mean Δ = 0.83 [0.78–0.95], *p* > 0.10, Figure 4f). 

## 4. Discussion

The effects of livestock grazing have generated extensive discussions [51,52]. The economic and socio-environmental benefits have been widely recognized [53]. However, many natural terrestrial areas are being altered by livestock, and livestock–wildlife conflict has become an increasingly widespread problem worldwide [8,54]. Our study provides insight into the effects of livestock on the spatial and temporal distribution patterns and behaviour of Reeves’s Pheasant. We tested the effects of anthropogenic disturbances and various natural environmental factors on the spatial and temporal distribution patterns of Reeves’s Pheasant at a fine spatial and temporal scale over one year. We found that the detection probability of Reeves’s Pheasant during the breeding season was positively correlated with distance from the nearest impervious road and negatively correlated with distance from the nearest settlement, while this trend did not occur during the non-breeding season. One possible explanation is that pheasants are more sensitive to human disturbance during the breeding season [22]. Some studies have also shown that wildlife will approach infrastructure to avoid natural predators [55] and move away from roads to avoid roadkill [56]. Reeves’s Pheasant did not avoid livestock as we expected. Instead, our analysis revealed a high degree of spatio-temporal overlap between livestock and Reeves’s Pheasant during both the breeding and non-breeding seasons, which is more likely due to their similar habitat preferences and possibly because the behavioural patterns of this bird were altered by livestock. Our results illustrate that integrating spatial and temporal patterns of wildlife distribution with behavioural observations can reveal important information on how wildlife responds to livestock grazing. Considering the potential negative impacts of livestock in nature reserves on wildlife, monitoring the effects of livestock activities on target species is essential for identifying potential short-term disturbances and medium- and long-term consequences [17,21].

Reeves’s Pheasant is an elusive species that is often difficult to observe in the wild [30]. Earlier studies have investigated habitat and population abundance mainly by using necklace radio transmitters or line transect sampling [22,27]. These studies describe the habitat selection and distribution of Reeves’s Pheasant. However, using necklace radio transmitters is expensive and invasive, and line transect sampling data contain little information. This study demonstrates the feasibility of using camera traps to study the habitat use of Reeves’s Pheasant.

The detection probability of Reeves’s Pheasant during the breeding season was higher in areas far from roads, which is in line with previous findings that road construction has a negative impact on wildlife at a large geographical scale [57,58]. At small geographical scales, roads severely affect survival [59] and gene exchange [60]. Although some studies suggest that high traffic volumes may be beneficial for the survival and reproduction of wildlife living in cities close to roads [61,62], in the forest, the negative impact of road networks on wildlife often outweighs the benefits [63] and influences the survival and habitat selection of wild animals [57]. Our results support the conclusions of previous studies [57,64].

Animals’ daily activity patterns are regulated by endogenous rhythms, which are shaped by external environment cues that change over a 24-h period [65]. We observed a consistent bimodal diel activity pattern of Reeves’s Pheasant across seasons. The activity pattern of Reeves’s Pheasant showed no response to livestock presence during the breeding season, non-breeding season, or all year round. The activity peak of Reeves’s Pheasant occurred in the morning and evening during both the breeding and non-breeding seasons. Reeves’s Pheasant changed their foraging time when livestock were present. Mammals are also known to change their foraging times to avoid livestock activity [17,18]. Livestock activity was most intense in the afternoon. Pheasants tend to feed in the morning at camera trap sites where livestock are present. In contrast, when livestock are absent, Reeves’ pheasants tend to forage in the evening. The foraging behaviour of pheasants seems to evade livestock in temporal. Changes in the timing of activity are often detrimental to animals [66,67]. These contrasting activity patterns suggest that livestock presence may create a “fearful” environment [18], reducing birds’ foraging efficiency [68]. 

Animals adjust their behaviour and choose different habitats to improve their survival, reproduction, and fitness [46,69]. At the spatial scale, our findings suggest that livestock presence changed the behavioural patterns of Reeves’s Pheasant to varying degrees. Reeves’s Pheasant showed high vigilance even under natural conditions, probably due to their nervous nature [30]. It is important to emphasize that livestock presence further increases the vigilance of Reeves’s Pheasant and may have implications for the survival and reproduction of these birds. 

Increased vigilance can help wildlife avoid predators and improve their chances of survival [70,71]. However, livestock are not predators of pheasants. Therefore, increased vigilance may result from a mismatch between risk perception and behavioural response [72]. Even where human activity is non-lethal, the impacts of human disturbance can be analogous to the effects of predation risk [72]. Disturbance can induce costly anti-predator behaviours [46,73,74] that can compromise individual fitness and influence population dynamics in other predator–prey systems, with implications for entire ecosystems [75,76]. Animals perceive and respond to risks associated with human activity and infrastructure, even in the absence of a true threat [77,78]. Livestock not only compete with wildlife for resources but also modify the environment (e.g., through grazing and trampling) and affect the habitat conditions of wildlife [18,79]. Our study provides further evidence that livestock grazing increases the vigilance behaviour of Reeves’s Pheasant. We found that the pheasant vigilance levels appeared to be decreasing when livestock grazing intensity was greater than 30. This may be the result of an insufficient sample size. There is therefore a need for further research into the ecological consequences of high livestock grazing intensities. In many ecosystems, livestock compete with wildlife species for limited food [4] and space [80] and destroy bird eggs [25], which threatens their survival [7,81]. Many studies have shown that wild animals are forced to adjust their spatial distribution, circadian rhythms [17,52], and behaviour to adapt to livestock grazing [21], which may result largely from a fear response to livestock, although this has yet to be demonstrated experimentally [82]. The impacts of fear are typically mediated by changes in individual behaviour [47], which may vary spatially or temporally with changes in the risk perception of wildlife [47,73]. More research is needed to quantify the relative contribution of livestock to creating a ‘landscape of fear’.

## 5. Conclusions

Although wildlife responds to livestock grazing in complex ways, the direct and indirect negative impacts of extensive livestock grazing on nature have been well documented. Our study showed that livestock presence altered the behavioural patterns of Reeves’s Pheasant but had no significant influences on its distribution patterns at spatial or temporal scales. Livestock have a negative impact on this bird, such as reduced foraging efficiency and increased vigilance. As a result, their growth rate decreases, which is not conducive to population stability and recovery. Where habitat modification is unavoidable, we recommend the integration of behavioural ecological principles into nature reserve conservation plans to facilitate animal behavioural adaptations, resource acquisition, reproduction, and dispersal. Reducing negative anthropogenic impacts on animal behaviour will be key to biodiversity conservation in an increasingly human-dominated world.

## Figures and Tables

**Figure 1 animals-12-02968-f001:**
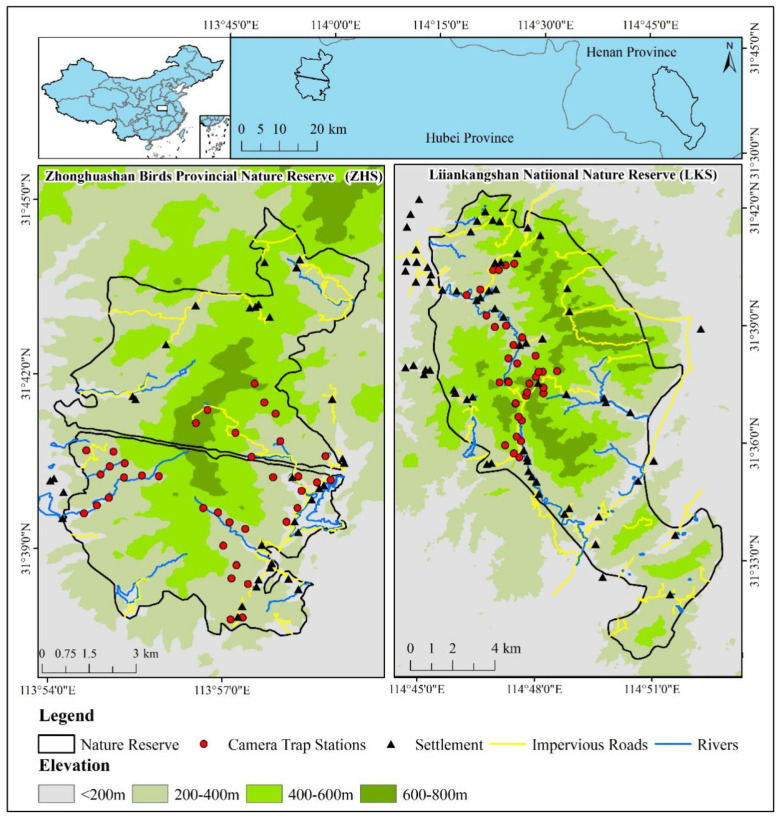
Camera trap stations in LKS and ZHS, inset showing the locations of the study areas in China.

**Figure 2 animals-12-02968-f002:**
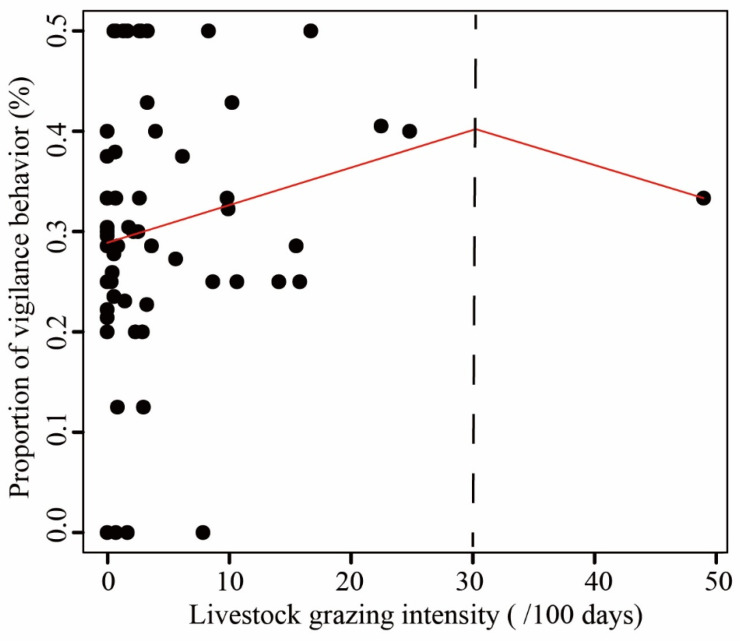
The relationship between livestock grazing intensity and the proportion of vigilance behaviour of Reeves’s Pheasant. A threshold in livestock grazing intensity is represented by the black-dashed line.

**Figure 3 animals-12-02968-f003:**
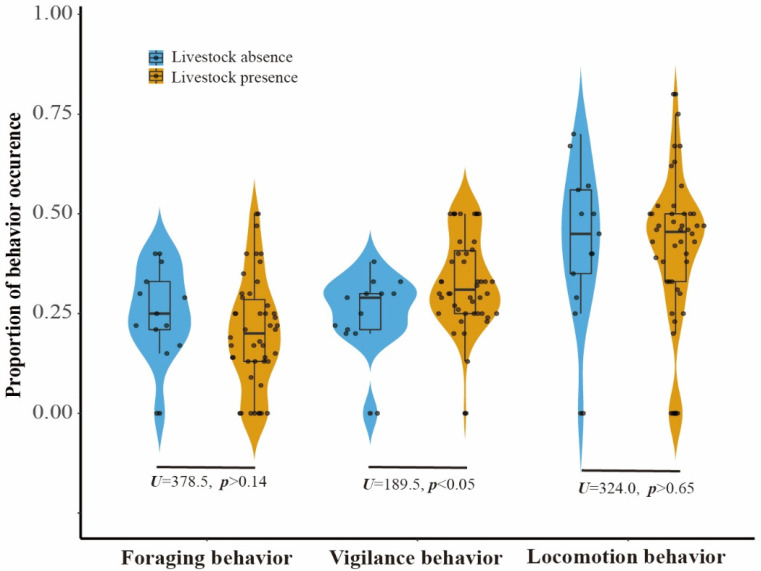
Proportion of Reeves’s Pheasant’s foraging, vigilance, and locomotion behaviour occurring at camera trap stations in the presence (yellow) and absence (blue) of livestock.

**Figure 4 animals-12-02968-f004:**
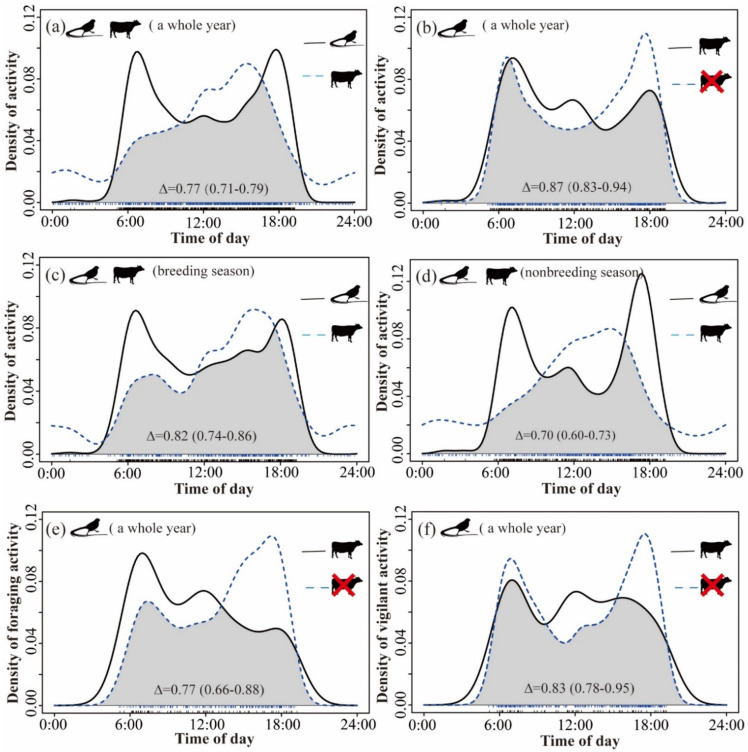
Activity pattern of Reeves’s Pheasant and livestock. Probability density distribution and overlap throughout the day for breeding season or nonbreeding season or the year in LKS and ZHS. Black silhouettes with long tails refer to Reeves’s Pheasant, and another to livestock. (**a**) Overlap in diel activity patterns of Reeves’s Pheasant and livestock throughout the year. The influence of livestock presence on the diel activity patterns of Reeves’s Pheasant (**b**). (**c**,**d**) The overlap of daily activity patterns of livestock and Reeves’s Pheasant in breeding season and non-breeding season, respectively. (**e**,**f**) The influence of livestock presence on foraging behaviour and vigilant behaviour pattern of Reeves’s Pheasant.

**Table 1 animals-12-02968-t001:** Sources of variables used for model analysis.

Covariates	Assessment	Description	Source
Vegetation type	Field measurements	Categorical	
Elevation	Field measurements	Numeric	
Distance from nearest river	ArcGIS 10.4	Numeric (m)	http://glovis.usgs.gov/ (accessed on 15 June 2019)
Distance from the nearest road	ArcGIS 10.4	Numeric (m)	http://glovis.usgs.gov/ (accessed on 15 June 2019)
Distance from the nearest settlement	ArcGIS 10.4	Numeric (m)	http://glovis.usgs.gov/ (accessed on 15 June 2019)
Distance from the nearest cultivated land	ArcGIS 10.4	Numeric (m)	http://glovis.usgs.gov/ (accessed on 15 June 2019)
Livestock encounter rates	camera trap data	Numeric	
Human encounter rates	camera trap data	Numeric	

**Table 2 animals-12-02968-t002:** Effect of *β*-coefficients and standard errors (SE) based on model averaging to assess occupancy and detectability of Reeves’s Pheasant.

Life History	Model Component	Covariates	*β*	SE	*p*	
Nonbreeding	Occupancy (*ψ*)	Intercept	1.60	0.55	0.004	**
season		Distance to the nearest impervious road	−0.28	0.38	0.458	
	Detection (*p*)	Intercept	−3.93	0.47	<0.001	***
		Location	0.91	0.19	<0.001	***
		Vegetation type	0.77	0.47	0.103	
		Elevation	0.02	0.06	0.720	
		Distance to the nearest settlement	−0.18	0.41	0.665	
		livestock encounter rates	0.08	0.49	0.869	
Breeding	Occupancy (*ψ*)	Intercept	1.55	0.46	<0.001	***
season		Location	0.52	0.72	0.470	
		Livestock encounter rates	1.13	1.27	0.373	
	Detection (*p*)	Intercept	−1.80	0.06	<0.001	***
		Distance to the nearest impervious roads	0.36	0.10	0.004	**
		Distance to the nearest settlement	−0.28	0.09	0.002	**

Significance codes: *** *p* < 0.001, ** *p* < 0.01.

## Data Availability

Raw data are available from the corresponding authors upon reasonable request.

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
