# Peer review of "Effects of Livestock Grazing on Spatio-Temporal Patterns and Behaviour of Reeves’s Pheasant *Syrmaticus reevesii"

_animals, 2022, doi:10.3390/ani12212968_

Round 1
Reviewer 1 Report
Animals 1967950
Reviewer’s comments
The authors have correctly identified behavioural change as an under-studied impact of livestock grazing upon wildlife, as opposed to better-studied distribution and spatial change. They have quite a good data set, from two sites, allowing robust statistical analysis. A feature I liked was the use of videos from camera traps, which permits the behavioural analysis that cannot be achieved merely with still photos that show presence/absence/frequency of occurrence. I was also pleased to see the very extensive data set in the supplementary material.
The suggestion that vigilance behaviour might decrease after livestock activity exceeds 30 (having risen significantly at lower but increasing livestock activity levels) seems to be based on only a single data point (Fig.2), and I would not attach much weight to this possibility – at least, not yet.
One of the important results is that foraging activity of the pheasants is reduced towards dusk, in those areas with livestock. The authors have noted the potential consequences for energetics (and hence survival). The finding seems to be statistically robust, yet it was not clear to me exactly how this relationship is caused. It is an interesting point worth further investigation. (e.g., why reduced foraging at dusk but not reduced at dawn?).
The text is very well written throughout, and the references are comprehensive and appropriate. I found all of the figures and tables well presented.
Reviewer 2 Report
The authors explored the effects of livestock grazing and human activities on the spatio-temporal distribution and behavioral patterns of Reeves's Pheasant. They found the presence of livestock did not affect the distribution of Reeves's Pheasant, but altered the behavior of this pheasant. The manuscript was well written, and adequately analysed. I recommend accept after minor revision.
Some results can be better presented.
Line 277-280, this result was tightly linked to the result in line 272-273, please re-organize this paragraph.
Line 306-308, this result was similar to line 294-298, may be authors can move these sentence a little bit. Focus on the same point in consecutive sentences.
The figure 4 was not very clear, perhaps the authors can improve it either by making the current figure more clear or presenting the results a different way.
Reviewer 3 Report
The Authors using camera-trap based data, try to empirical evidence that livestock presence influences pheasant behaviour. It is one of the innovative approach and inference drawn is very useful for management implications. Therefore, the publication this manuscript help management and thus conservation, besides getting its credit to the journal. Going by the current status of the manuscript, it cannot be suggested to be published in the present condition, as it is lacking clarity from the introduction to discussion. Firstly, the authors need to point out clearly what is all the objectives, which is not put forth at the introduction at present. Secondly, the method section is sketchy, and it should be elaborate for anyone to repeat the study elsewhere. Methods and results section should described/interpreted, as per the objective order, so that it easy for the audience to understand. Finally, there self-contradiction in some statements and interpretation, which needs to be eliminated carefully. All the above-mentioned lacunae are pointed out one-by-one below and if these comments and suggestions are incorporated, the manuscript improve in its quality and clarity.
Line No. 9-12: Previous studies have demonstrated the impact of livestock grazing and human presence on habitat use and movement patterns of birds, whereas little is known about the effect of free-ranging livestock on bird behavior. There is not much difference between the first part of the sentence [i.e., livestock grazing and human presence on habitat use and movement patterns of birds] and second part of the sentence [effect of free-ranging livestock on bird behavior].
Line No. 16-19: Livestock presence did not affect the spatio-temporal distribution patterns of Reeves's Pheasant but altered the behavior of this pheasant. Reeves’s Pheasant displayed high levels of vigilance during the day in areas with livestock, and its vigilance increased significantly with increasing intensities of livestock activity.
Modify the above as follows: Livestock presence though not affected the spatiotemporal distribution of the pheasant; its intensity significantly increased the time spent on vigilance by pheasant indicating the behavioural alteration.
Line No. 20-22: In addition, Reeves’s Pheasant mainly foraged in the early morning in areas with livestock, while foraged at dusk in areas without livestock.
Modify the above as: In addition, pheasants in areas with livestock foraged mostly during early morning, while in areas without livestock foraged at dusk.
Line No. 21-22: Therefore, free-ranging livestock can affect the behavior, especially vigilance and foraging behavior of Reeves’s Pheasant, which may affect the energy allocation of individual
Modify the above as: Therefore, the study concludes that livestock intensity in the nature reserve reduces the fitness of pheasants through increased vigilance and decreased feeding behaviours.
However, to make the above statement, the study needs to show empirical evidence that the feeding time of pheasants in areas with livestock is statistically lower compared to that of in areas without livestock.
Line No. 72-73: The aim of our study was to provide science-based conservation strategies for a regional landscape-scale recovery plan for Reeves’s Pheasant to resolve human-wildlife conflicts. Here what the authors saying aim is not actually aim of the study goal and it is the goal of the study. Further, in Line No. 76-81, the authors talk about how various data set have been analyzed and hence it should be moved to methodology section as part of analysis. Therefore, the authors have not mentioned what are the objectives of the study and the introduction should end with the same clearly. The result section should follow one-by-one the objectives.
Line No. 126: We tested a suite of natural and anthropogenic variables that could affect the distribution… Replace the word ‘suite’ with the word ‘set’
Line No. 128-131: We investigated the effect of vegetation type, elevation, distance from the nearest 128 river, distance from the nearest road, distance from the nearest settlement, distance from 129 the nearest cultivated land, livestock encounter rates, and human encounter rates on the occupancy of Reeves's Pheasant.
If the If the authors mean to say ‘a suite of natural and anthropogenic variables’ are the variables listed in the line no. 128-131 lines, I suggest the sentence could be merged with line no. 126.
Line No. 131-133: Environmental variables were measured in the field or …..downloaded loaded from open access sources (DEM data from the Geospatial Data Cloud and Remote sensing images data from USUG).
The above statement could be changed as ‘The variables listed were assessed at the respective camera-trap locations either directly or downloaded from loaded from open access sources (DEM data from the Geospatial Data Cloud and Remote sensing images data from USUG).
Further, the authors are suggested to come out with a table showing (i) What all the variables were assessed in the field (ii) How they were assessed? Giving a detailed methodology including instrument and their model or software and the version used. (iii) What is all the variables were downloaded from open access sources. Methodology for some variables is described below in the text. Instead of text, if these methodologies are given in a table, it is easy for every one understand.
Line No. 147: arity, and variables with r > 0.7 were not used in the same model (Dormann et al. 2013). Please delete the word same in this line.
Line No. 166: we measured the distance from each site in the presence of Reeves’s Pheasant to the…..
What the authors mean site in the above sentence and why in the presence of pheasant the variable were assessed. I thing for each camera trap location the variables were assessed, if so modify. Secondly, the variables need not be or cannot be assessed in presence of pheasant. All variable should have been assessed irrespective of the study subject was recorded/detected or not by a camera trap in its location. So that how detection probability varies with each variable could be assessed. Please rewrite the same clearly.
Line No. 170-172: River is an important resource for wildlife to maintain normal activities, and differences in vegetation type and elevation have been reported to have important effects on the habitat use of target species (Xu et al. 2009; Tian et al. 2020). This kind of information should be at the introduction and here in the methodology.
Results
Line No. 246-255: These are not actually results and are part of methodology stating the sample size and hence they can be placed at the end of methodology (after Line No. 244) as a separate paragraph.
3.1. Spatial distribution of Reeves’s Pheasant and livestock
Line No. 258-260: The occupancy estimation model included both location and livestock encounter rates, but the 95% confidence interval for the estimate parameter (β) included 0 (Table S5). Here apart from repeating the information what is there in the table in the form text, authors must mention, what it means i.e. occupancy model including the location and the livestock encounter means what needs to explained. Likewise, but the 95% confidence interval for the estimate parameter (β) included 0 indicate what needed to be explained. Draying inference from the table or figure is most essential in result text, which is called interpretation. Although the paragraph in Line No. 257-266 deals with spatial distribution of pheasant and livestock, no meaningful inference is drawn out of this data analysis in this paragraph.
Figure 2. In this figure the time spent on vigilance at >40 livestock intensity seems like an outlier and it is just based on one observation. Is this right to include the outlier in the analyses? What the trend if you exclude the outlier. I hope it may better explain without than with.
3.2. Behavioral patterns of Reeves's Pheasant:
Under this heading the authors talk about three behaviours viz. locomotion, vigilance and foraging. These behaviours needed to described as what all the activities brought under what behaviour need to be described or given as ethogram.
Similarly, there are quantitative measures of data showing density of activities. What the density of activities means, how it was assessed and in what unit it is worked out? Are not at all mentioned in the method section.
As mentioned above what the authors mean by foraging density? And how it was arrived needs to described at the methods section.
Line No. 274-276: The effect of livestock on pheasant vigilance behavior appeared to diminish at livestock activity intensities greater than 30, but the proportion of vigilance behavior occurring was still higher than livestock absence station (Fig. 2).
The second part of the above sentence is contradicting the first part of the sentience. And the same way when proportion of vigilance behaviour is higher in livestock absent areas compared to present areas, how the probability of vigilance behavior of pheasant was significantly higher in areas with livestock than in areas without livestock as mentioned in 278-279 & shown in Fig. 3.
Figure 3. Proportion of Reeves's Pheasant’s foraging, vigilance and locomotion behavior occurring at camera trap stations in the presence (yellow) and absence (blue) of livestock. When the available total time for pheasant is constant irrespective livestock presence or absence areas and in the livestock presence areas when pheasant spent significantly more time on vigilance, it should reduce time spent on any other activity/ies viz. feeding or and locomotion, and thereby any one of the two activities should also vary significantly compared to absence areas. How the authors saying vigilance alone varied but other activities or not varied?
Discussion
Line No. 331-333: detection probability of Reeves’s Pheasant during the breeding season was negatively correlated with distance from the nearest impervious road and positively correlated with distance from the nearest settlement, while this trend did not occur during the non-breeding season
In the above lines the reason why distance from the nearest impervious road is negatively influenced and why distance from the nearest settlement positively influenced the detection probability of pheasant during breeding season. Why these are not influenced during non-breeding season. These need justifications to make these findings as conclusions.
Line No. 346-348: Earlier studies have investigated habitat and population abundance mainly by using necklace radio transmitters or line transect sampling (Xu et al. 2009; Zhou et al. 2015). These studies describe the habitat selection and distribution of Reeves's Pheasant, but the data contain little information and are costly to come by.
I agree with the above opinion of the authors with regards to radio-collaring study but not with line-transect sampling. Because line transect survey using single observer cannot be expensive compared to camera trap-based study cost. Multi-Covariate Distance Sampling with low cost would likely to yield in-depth information about the influence of various covariates on the detection probability and density than camera-trap based data. So I suggest the authors to modify the above sentence accordingly.
In Line No. 331: the detection probability of Reeves’s Pheasant during the breeding season was negatively correlated with distance from the nearest impervious road. But in the line No. 351, The detection probability of Reeves’s Pheasant during the breeding season was higher in areas far from roads. These two are contradicting. As per line no. 331, negative correlation means, the detection probability decreases (dependent factor) with increase in distance from nearest road (independent factor/covariate), but the Line No. 351 is interpreting as positive influence or correlation. Please confirm these and correct where you went wrong. Distance from road is different from distance to road? In the former distance is calculated from the road to location where pheasant detected, while in the later distance is calculated from the pheasant detection site to the road. Accordingly, the positive and negative effect changes.
Line No. 372: On the contrary, in areas where livestock were absent, Reeves’s Pheasant preferred to forage in the afternoon. Authors have to be cautious about the time in the above line Pheasant preferred to forage in the afternoon, which is also a period in which livestock activity is most intensive. Is that the findings of the study.
Line No. 414: This bird may impose costs on the non-lethal effects of livestock. This is not clear, please rewrite it.
